# Simvastatin Induces Unfolded Protein Response and Enhances Temozolomide-Induced Cell Death in Glioblastoma Cells

**DOI:** 10.3390/cells9112339

**Published:** 2020-10-22

**Authors:** Sanaz Dastghaib, Shahla Shojaei, Zohreh Mostafavi-Pour, Pawan Sharma, John B. Patterson, Afshin Samali, Pooneh Mokarram, Saeid Ghavami

**Affiliations:** 1Department of Biochemistry, School of Medicine, Shiraz University of Medical Sciences, Shiraz 7134845794, Iran; suny.respina@gmail.com (S.D.); zmostafavipour88@yahoo.co.uk (Z.M.-P.); 2Endocrinology and Metabolism Research Center, Nemazee Hospital, Shiraz University of Medical Sciences, Shiraz 7193635899, Iran; 3Department of Human Anatomy and Cell Science, Rady Faculty of Health Sciences, Max Rady College of Medicine, University of Manitoba, Winnipeg, MB R3E 0J9, Canada; Shahla.Shojaei@umanitoba.ca; 4Maternal-Fetal Medicine Research Center, School of Medicine, Shiraz University of Medical Sciences, Shiraz 7134845794, Iran; 5Center for Translational Medicine, Division of Pulmonary, Allergy and Critical Care Medicine, Jane & Leonard Korman Respiratory Institute, Sidney Kimmel Medical College, Thomas Jefferson University, Philadelphia, PA 19107, USA; pawan.sharma@jefferson.edu; 6Orinove, Newbury Park, CA 91320, USA; John.Patterson@orinove.com; 7Apoptosis Research Centre, National University of Ireland, H91 W2TY Galway, Ireland; afshin.samali@nuigalway.ie; 8Autophagy Research Center, Shiraz University of Medical Sciences, Shiraz 7134845794, Iran; 9Colorectal Research Center, Shiraz University of Medical Sciences, Shiraz 7193635899, Iran; 10Faculty of Medicine, Katowice School of Technology, 40-555 Katowice, Poland

**Keywords:** statin, glioblastoma, ER stress, mevalonate cascade, autophagy, autophagy flux

## Abstract

Glioblastoma (GBM) is the most prevalent malignant primary brain tumor with a very poor survival rate. Temozolomide (TMZ) is the common chemotherapeutic agent used for GBM treatment. We recently demonstrated that simvastatin (Simva) increases TMZ-induced apoptosis via the inhibition of autophagic flux in GBM cells. Considering the role of the unfolded protein response (UPR) pathway in the regulation of autophagy, we investigated the involvement of UPR in Simva–TMZ-induced cell death by utilizing highly selective IRE1 RNase activity inhibitor MKC8866, PERK inhibitor GSK-2606414 (PERKi), and eIF2α inhibitor salubrinal. Simva–TMZ treatment decreased the viability of GBM cells and significantly increased apoptotic cell death when compared to TMZ or Simva alone. Simva–TMZ induced both UPR, as determined by an increase in GRP78, XBP splicing, eukaryote initiation factor 2α (eIF2α) phosphorylation, and inhibited autophagic flux (accumulation of LC3β-II and inhibition of p62 degradation). IRE1 RNase inhibition did not affect Simva–TMZ-induced cell death, but it significantly induced p62 degradation and increased the microtubule-associated proteins light chain 3 (LC3)β-II/LC3β-I ratio in U87 cells, while salubrinal did not affect the Simva–TMZ induced cytotoxicity of GBM cells. In contrast, protein kinase RNA-like endoplasmic reticulum kinase (PERK) inhibition significantly increased Simva–TMZ-induced cell death in U87 cells. Interestingly, whereas PERK inhibition induced p62 accumulation in both GBM cell lines, it differentially affected the LC3β-II/LC3β-I ratio in U87 (decrease) and U251 (increase) cells. Simvastatin sensitizes GBM cells to TMZ-induced cell death via a mechanism that involves autophagy and UPR pathways. More specifically, our results imply that the IRE1 and PERK signaling arms of the UPR regulate Simva–TMZ-mediated autophagy flux inhibition in U251 and U87 GBM cells.

## 1. Introduction

Glioblastoma multiforme (GBM) is the most common malignant tumor of the central nervous system (CNS) [1,2,3]. The tumor is invasive and incurable with poor prognosis and inevitable recurrences [4,5]. The average survival of GBM patients is about 15 months [6,7]. GBM therapy comprises surgery, temozolomide (TMZ), radiochemotherapy, and TMZ maintenance chemotherapy [8]. TMZ is administered orally and induces apoptotic cell death due to DNA damage [9]. 

Statins are FDA-approved mevalonate (MEV) cascade inhibitors that directly inhibit 3-hydroxy-3-methyl-glutaryl–CoA (HMG–CoA) reductase [10,11]. They reduce cholesterol synthesis by interfering with the signaling pathways involved in tumorigenesis including intermediates that are involved in the prenylation small Rho GTPases [12]. Epidemiological studies have shown that the long-term consumption of statins reduces mortality rates of different types of cancer, including GBM (it is worth mentioning that in GBM, it reduces the mortality but is not statistically significant) [13,14]. In addition, preclinical studies have confirmed the anticancer, anti-proliferative, and anti-metastatic properties of statins in several cancers such as myeloma, leukemia, glioma, breast, and prostate cancer [15,16,17]. 

Autophagy, apoptosis, and the unfolded protein response (UPR) play critical roles in determining the fate of cancer cells before and after chemotherapy [18,19]. These mechanisms play an important role in the regulation of TMZ chemo-resistance in GBM [20,21,22]. GBM tumor progression is associated with increased endoplasmic reticulum (ER) stress due to the high demand for protein folding and intercellular transport of newly synthesized proteins [23]. Tumor cells utilize several signaling pathways to restore cellular homeostasis or respond to chemotherapy induced-stress, including macroautophagy (hereafter autophagy), UPR, ER-associated protein degradation (ERAD), and apoptosis [20]. 

Autophagy degrades damaged organelles and allows misfolded proteins to be recycled in response to cellular stress and starvation, thereby suppressing tumor development in the early stages of cancers, whereas it improves tumor progression at advanced stages of tumor development via inducing resistance to chemotherapeutic agents [24,25,26,27,28]. The UPR is also activated in response to the accumulation of unfolded proteins in the ER [29]. The UPR is orchestrated by three ER transmembrane proteins, activated transcription factor 6 (ATF6), protein kinase RNA-like ER kinase (PERK), and serine/threonine-protein kinase/endoribonuclease inositol-requiring enzyme 1α (IRE1α) [23,30]. Upon activation, UPR signaling may result in cell adaptation to stress or cell death, depending on cell type, nature of the stress stimulus, and duration and severity of cell stress [21,31]. The UPR has emerged as a therapeutic target for the treatment of cancer, as this pathway is over-activated in cancer in comparison to healthy, non-proliferative cells [32]. UPR is also involved in the shaping of tumor microenvironment through an IRE1α-dependent alteration of the tumor cells secretome [33]. As such, targeting IRE1α could serve as a potential therapeutic option in cancers. 

We recently reported a close link between autophagy and UPR [34,35,36,37]. Others have also reported crosstalk between autophagy and UPR in different in vitro and in vivo models. Activation of the IRE1 arm regulates autophagy by modulating Beclin-1 protein expression [38,39], whereas PERK induces the expression of several proteins belonging to the autophagy-related genes (ATG) family, including ATG 3, 5, 7, 10, and 12 [20]. 

We have recently demonstrated that Simvastatin (Simva) increases TMZ-induced apoptosis via targeting autophagosome/lysosome fusion in both GBM cell lines and primary GBM patient-derived cells [40,41]. Considering the role of the UPR pathway in the regulation of autophagy and cell death pathways [27,36,37,42], our hypothesis was to investigate how UPR is involved in the regulation of Simva–TMZ combination therapy cell death in GBM cells. For this purpose, we evaluated the effects of Simva–TMZ on the UPR arms (IRE-1α and PERK) in GBM cells, using highly selective IRE-1α and PERK inhibitors. We later assessed how these inhibitors were involved in the regulation of Simva–TMZ effects on autophagy flux and cell death in GBM cells. 

## 2. Results 

### 2.1. Co-Treatment with Simva-TMZ Increases Mevalonate Caspase Independet Cytotoxicity and Autophagy Flux Inhibtion in GBM Cells

In our previous investigations, we have shown that Simva–TMZ combination treatment significantly reduced cell viability in U87 and U251 as compared with Simva and TMZ alone [41]. Briefly, we treated GBM cells (U87 and U251) with TMZ (100 µM), Simva (1 µM for U251 and 2.5 µM for U87), and mevalonate (2.5 mM) at the 72 h time-point for all of our treatments. We chose these concentrations based on the initial screening of the cell viability experiments for different concentrations of these compounds on GBM cells (Appendix A). We selected Simva and the TMZ concentrations in each cell line based on their cell viability in 72 h (70% < cell viability < 95%), so the single treatment has toxicity less than 30%. We also selected mevalonate concentration based on its inhibitory effect on mevalonate cascade and the lowest cytotoxic effects on GBM cells, as previously established by our team [40,41,43]. 

Later, we further confirmed that Simva–TMZ co-treatment significantly reduced cell viability in U87 and U251 as compared with Simva (*p* < 0.05 and *p* < 0.0001, respectively) and TMZ alone (*p* < 0.01 and *p* < 0.0001, respectively) (Appendix A). Furthermore, co-treatment with mevalonate did not have any significant effects on the loss of cell viability induced by Simva–TMZ (Appendix A). 

We further confirmed that co-treatment with Simva and TMZ induced significant apoptosis in U87 and U251 cells (22.6% and 34.3%, *p* < 0.00001) compared with TMZ (7.74% and 21.5%) and Simva alone (15.7% and 29%) at 72 h, as determined by changes in the sub-G1 DNA content of the cells (Appendix A). Our observations showed that TMZ single treatment induced significantly lower apoptosis compared to Simva. Single treatment in both cell lines reflects the different apoptosis mechanisms of these compounds [40,41]. In both cell lines, mevalonate (2.5 mM) prevented apoptosis induced by Simva alone and partially inhibited Simva–TMZ-induced apoptosis. It is worth mentioning that the inhibitory effect of mevalonate on Simva–TMZ-induced apoptosis was far more pronounced in U87 than in U251 cells. These findings suggest that Simva–TMZ-induced apoptosis is partially dependent on mevalonate cascade. 

We also assessed the protein amount of the autophagy markers, Beclin-1, p62, and microtubule-associated proteins light chain 3 (LC3β)-II in U87 and U251 treated cells at 72 h (Appendix A). Co-treatment with Simva and TMZ increased Beclin-1 protein amount and induced the accumulation of p62 and lipidated LC3β-II compared to the time-matched control. Autophagy flux refers to the flow of the autophagosome to lysosomes for degradation and usually is detected using LC3-β lipidation and the degradation of p62 [44]. These findings confirm our recent findings that Simva–TMZ inhibits autophagic flux in GBM cells [41]. 

### 2.2. Co-Treatment with Simva–TMZ Induces UPR in GBM Cells

We also studied the protein amount of GRP78, IRE1, XBP1s, ATF6, eIF2α, and phospho-eIF2α (p-eIF2α) proteins, which are key markers of the UPR pathway in the GBM cell models upon treatment with TMZ, Simva, and Simva-TMZ, after 72 h. We also used dithiothreitol (DTT, 10 mM) as a positive control for UPR induction. We also used time zero control for all experiments to evaluate the status of UPR at the beginning of the treatments. In both cell lines, Simva–TMZ significantly increased GRP78, IRE1, XBP-1s, and ATF-6 protein amount compared to treatment with Simva or TMZ alone. Simva–TMZ co-treatment significantly increased the p-eIF2α/eIF2α ratio in U87, while it decreased in U251 compared to Simva and TMZ single treatment (Figure 1, Appendix A). In addition, Simva–TMZ significantly induced caspase-3 cleavage in parallel to UPR induction in both cell lines (Appendix A). It is worth mentioning that some proteins (e.g., ATF6) showed different immunoblotting patterns possibly due to various isoforms or a possible phosphorylated site between U87 and U251 cells.

### 2.3. IRE-1α RNase Inhibition Does Not Impact the Effects of Simva, TMZ, or Simva–TMZ on Cell Viability of GBM Cells

We treated GBM cells with different doses of the selective IRE-1α RNase inhibitor, MKC8866 (10–80 µM) for 48 and 72 h (Figure 2A,B). MKC8866 decreased the viability of U251 and U87 cells (80 µM, *p* < 0.0001 and 30 µM, *p* < 0.001, respectively) at 72 h compared to the time-matched control. In addition, our results showed that MKC8866 (30 µM) inhibited IRE1-1α RNase activity as determined by a decrease in XBP-1s protein levels upon treatment with Simva–TMZ (Figure 3A–C) and by qPCR upon treatment with Simva or Simva–TMZ (Figure 3D,E) in both cell lines. The treatment of cells with either Simva or TMZ alone moderately reduced XBP-1 mRNA expression in U87 cells, which was further reduced for Simva but normalized to control levels for TMZ by MKC8866.

Next, we evaluated if the IRE-1/XBP-1 axis is involved in the Simva-mediated sensitization of GBM cells to TMZ-induced cell death at 72 h to correlate investigate the possible regulation of autophagy flux via IRE-1/XBP-1. We inhibited IRE-1 RNase activity using MKC8866 (30 µM) and treated U87 and U251 cells with Simva, TMZ, or Simva–TMZ for 72 h. As shown in Figure 2C,D, IRE-1α RNase inhibition did not affect Simva-, TMZ-, or Simva–TMZ-induced cell death in either cell line. As shown in Figure 2E,F, IRE-1α RNase inhibition did not affect Simva-, TMZ-, or Simva–TMZ-induced apoptosis in either cell line. 

### 2.4. Inhibition of IRE-1α RNase Activity Regulates the Effect of Co-Treatment with Simva and TMZ on Autophagic Flux Inhibition without Affecting the Overall Cytotoxic Effect

Co-treatment with Simva and TMZ reduced autophagy flux in U87 and U251 cells (as shown in Appendix A), and MKC8866 inhibited these effects (Figure 4A–G). In particular, MKC8866 dramatically reduced the level of Beclin-1 and p62 induced by Simva–TMZ in both cell lines (*p* < 0.0001, Figure 4A–G), suggesting that degradation of the autophagosome is regulated by the IRE1–XBP1s pathway. However, MKC8866 did not affect the Simva–TMZ-induced increase in the LC3-βII/LC3-βI ratio in U87 cells, while it significantly increased the ratio in U251 cells in the presence of Simva. These findings indicate that in the presence of Simva–TMZ, IRE1-1-XBPs inhibition does not change the production of autophagosomes in U87 cells, but it increases autophagosome formation in U251 cells. As MKC8866 did not affect Simva–TMZ-induced cell death in U87 and U251 cells, we concluded that XBP-1 splicing is involved in the Simva–TMZ-mediated inhibition of autophagy flux, but it does not affect the cytotoxic effects of Simva–TMZ. This means that the IRE1/XBP-1 arm is functionally insignificant in the cytotoxic effects of Simva–TMZ in GBM cells.

### 2.5. PERK Inhibition Regulates the Effect of Co-Treatment with Simva–TMZ on Autophagy Flux in U87 and U251 Cells and Increased Simva–TMZ-Induced Cell Death in U87 Cells

With the exception of the lowest dose tested (1 μM) in U87 cells, incubation with the PERK inhibitor GSK-2606414 (PERKi) (1–20 µM) for 72 h had significant cytotoxic effects on GBM cells (Figure 5A,B). Based on these results, 5 μM was used for further experimentation as it is the lowest concentration inducing a near-maximum effect on cell viability. 

We carried out a dose–response of PERKi (1–20 µM) for 72 h and selected 5 μM for further experiments. We pretreated cells with PERKi (5 µM, 30 min) and then co-treated with Simva, TMZ, and Simva–TMZ for 48 and 72 h. PERK inhibition did not change the cytotoxic effect of Simva–TMZ in U251 cells, whereas it further reduced cell viability in Simva–TMZ-treated U87 cells (Figure 5C,D). Our apoptosis assay showed that Simva–TMZ co-treatment with PERKi did not significantly change Simva–TMZ-induced apoptosis (Figure 5E,F). Immunoblotting data revealed that the PERKi blocked the basal kinase activity of PERK and suppressed eIF2α phosphorylation (*p* < 0.0001). Co-treatment with Simva–TMZ by itself significantly reduced the p-eIF2α/eIF2α ratio, which was not further decreased by the PERKi in either cell line (Figure 6A–C).

We also investigated the role of PERK in the Simva–TMZ-mediated inhibition of autophagy flux in U87 and U251 cells (Figure 7). The PERKi significantly inhibited p62 degradation (*p* < 0.0001) in both cell lines, while the LC3β-II/LC3β-I ratio was decreased in U87 (*p* < 0.0001) and increased in U251 (*p* < 0.0001) cells. These findings indicate that PERKi decreased autophagosome formation (LC3β-II/LC3β-I) in U87 cells, induced autophagosome formation (more flux inhibition) in U251 cells, increased Simva–TMZ-induced cell death in U87 cells; thus, we conclude that the PERKi effect on Simva–TMZ-induced cell death might be related to autophagic flux.

### 2.6. Phospho-eIF2α Phosphatase Inhibitor Did Not Change Simva-TMZ-Induced Cell Death in U87 and U251 Cells

Treatment of U87 and U251 cells with the phospho-eIF2α phosphatase inhibitor (salubrinal; 1–20 µM), for 48 and 72 h, had significant cytotoxicity (except for 1 and 5 µM for 48 h in U87 cells, Figure 8A,B). Next, cells were pretreated with salubrinal (15 µM) for 30 min before treatment with TMZ, Simva, or Simva–TMZ. There was no detectable effect of salubrinal on the cytotoxicity of any of the co-treatments (Figure 8C,D). We also measured apoptosis, and the cytotoxic results were confirmed (Figure 8E,F). Immunoblotting data revealed that in combination with Simva–TMZ, salubrinal increased eIF2α phosphorylation and the p-eIF2/eIF2α ratio (*p* < 0.0001) in both cell lines (Figure 9A–C).

## 3. Discussion

In the present study, we showed that co-treatment of Simva with TMZ (Simva–TMZ) increased the sensitivity of GBM cells to TMZ-induced cell death, which occurred concomitantly with UPR induction and autophagic flux inhibition. According to our current findings, Simva–TMZ significantly decreased cell viability in a mevalonate independent fashion and increased apoptotic cell death, which could (only) partially be inhibited by mevalonate. Therefore, we conclude that Simva probably sensitizes GBM cells to TMZ-induced cell death through cellular pathways independent of HMG–CoA inhibition and the mevalonate cascade. 

The results of the MTT and the apoptotic flow cytometry assays for Mev–Simva–TMZ were not consistent. Based on MTT data, the cytotoxic effect of Simva–TMZ could not be restored by mevalonate, while propidium iodide (PI) analysis revealed that mevalonate could partially inhibit the effect of combined treatment. This controversy points to some limitations of the MTT method. First, although the tetrazolium-based MTT assay is a standard cell viability assay, reports are indicating the interference of such components during the staining process; second, the enzymatic conversion of formazan crystals is (merely) sensitive to metabolic rate [45,46,47]. Therefore, the findings of the current study suggested that co-treatment with Simva and TMZ induced cell death through pathways partially dependent on mevalonate cascade; however, pleiotropic mechanisms beyond the mevalonate cascade are likely more dominantly involved in the cytotoxic effects of Simva–TMZ. 

UPR is a defense mechanism against misfolded and aggregated proteins [48] and is orchestrated by three ER membrane sensors, ATF6, PERK, and IRE-1, which become inactive by binding to GRP78 (Bip). UPR is connected to apoptotic cell death via different intermediates, including C/EBP homologous protein (CHOP), phorbol-12-myristate-13-acetate-induced protein 1 (NOXA), c-Jun N-terminal kinase (JNK), and caspase-4 and caspase-12 [36,49,50]. Autophagy is also responsible for the degradation and removal of damaged organelles and misfolded proteins [51,52,53]. The term autophagic flux refers to the degradation activity of the autophagy pathway through the formation of the phagophore and engulfment of its p62-tagged cargo inside to form autophagosomes [41,54,55]. Autophagy and UPR pathways are similar to double-edged swords in anticancer treatment; low levels of activation are protective, while higher levels augment cell coping mechanisms and induce cell death [20,56,57,58].

Previously, we showed that Simva–TMZ induced cell death in GBM cells by autophagic flux inhibition in a cholesterol-independent manner, since the addition of mevalonate, cholesterol, farnesyl pyrophosphate, or geranylgeranyl pyrophosphate had no significant effects [41]. In the current study, co-treatment with Simva–TMZ induced UPR and inhibited autophagy flux in U87 and U251 cells. It increased the GRP78, ATF-6, and IRE-1 protein amount, as well as the XBP-1s (mRNA and protein) level, while it decreased the p-eIF2α/eIF2α ratio downstream of phospho-PERK, compared with the time-matched controls. 

It has been suggested that the UPR can be exploited in two ways to target cancer cells [22,59]: (1) target each arm of UPR individually to induce cell death directly or (2) exacerbate the whole cascade to make cells (more) susceptible to chemotherapies. There are some contradictory reports on the effects of the statins on UPR. It can inhibit [60] or activate [12] UPR and ER stress, whereas other studies report no effects at all of the statins on these pathways [61]. Evidently, it appears that statins exert differential ER stress responses, depending on cell type and the nature, magnitude, and duration of stress [36]. Our observation that Simva–TMZ treatment induced an increase in the protein amount of IRE-1, ATF6, and PERK (indicative of UPR change) is in line with research in mouse macrophage RAW264.7 cells and bone marrow-derived macrophages (BMDM), suggesting that statins (lovastatin and fluvastatin) upregulate the canonical UPR targets GRP78, ATF6, and XBP1, but not ATF4 and CHOP [12]. In addition, in human atrial fibroblasts, Simva increases the level of IRE-1, BiP/GRP78, and cleavage of ATF6, phosphorylation of PERK, XBP1 splicing, and nuclear accumulation of ATF4 and ATF6 [12]. 

To investigate the mechanisms of cell death and determine which signaling arm(s) of the UPR were most affected, we used pharmacological inhibitors of PERK (PERKi), IRE-1α RNase (MKC8866), and p-eIF2α phosphatase (salubrinal). In the presence and absence of these inhibitors, we evaluated the effects of Simva–TMZ on cell viability and the protein levels of downstream transcription factors (XBP-1s and p-eIF2α/eIF2α ratio). Our findings revealed that Simva–TMZ induced the IRE-1 signaling arm and increased the level of XBP-1s, while MKC8866 failed to prevent or inhibit the cytotoxic effect of Simva–TMZ. These findings indicate that other possible mechanisms are involved in Simva–TMZ-induced cell death in GBM cells and that although IRE-1α RNase inhibition modulates Simva–TMZ-induced signaling, it does not affect the overall cytotoxic effects. 

Confirming findings from previous studies, we showed that Simva–TMZ inhibited autophagy flux [41,62,63]; we demonstrate here for the first time that this is dependent on the IRE1–XBP-1s axis (as MKC-8866 inhibited the Simva–TMZ effect). To our surprise, the cytotoxic effect of Simva–TMZ was not affected by MKC8866. In our previous work, we found that Simva–TMZ likely triggered GBM cell death via the inhibition of autophagy flux; however, our current findings show that the prevention of autophagy flux inhibition using MKC8866 was not able to inhibit Simva–TMZ-induced cell death in GBM cells. A recent study showed that Coxsackievirus B3 autophagy induction in HeLa cells is dependent on UPR arms as an inhibition of PERK, IRE1, or ATF6 significantly decreased autophagy induction [64]. Researchers also showed that kaempferol (a flavonoid) induced autophagy cell death through the IRE1 pathway in gastric cancer [65]. Caffeine induces autophagy in hepatic stellate cells via IRE1 induction, and IRE1 knockdown in these cells decreases caffeine-induced autophagy flux [66]. Furthermore, UPR transmembrane receptors, PERK and IRE1, modulate p62 gene regulation in HCT116 cells [67], and IRE1 is linked to autophagy regulation after tunicamycin-induced ER stress in MCF-7 and MDA-MB-231 cell lines [68]. Interestingly, Huntington’s disease has been associated with autophagy flux inhibition via IRE1 activation, demonstrating the role of IRE1 in the regulation of autophagy flux in neural cells [69]. Overall, we can conclude that the regulation of autophagy (flux) via IRE1 highly depends on stimulus and cell type. In addition, it appears that even though IRE1 regulates autophagy flux, it might not contribute to the overall functional outcome, defining it as more of a fine-tuning mechanism. Whether this calls into question the functional contribution of autophagy flux to cell death in GBM cells warrants further investigation.

The PERKi did not change Simva–TMZ-induced cell death in U251 cells, but it increased Simva–TMZ cytotoxicity in U87 cells and decreased p62 degradation in both cell lines. In addition, the PERKi increased the LC3-β-II/LC3-β-I ratio in U251 cells but decreased it in U87 cells. These findings show that PERK differentially regulates autophagy flux in U251 and U87 cells. Although its role in autophagosome degradation appears similar in both cell lines, its contribution to autophagosome formation is clearly different. Our results imply that the Simva–TMZ-mediated inhibition of autophagy flux and (subsequent) induction of cell death is connected to the PERK pathway in U87 cells but not in U251 cells. Recently, it has been shown that apoptosis and autophagy induced by sterol regulatory element-binding proteins are related to PERK and UPR in osteosarcoma cells [70]. Therefore, it is possible that in U87 cells, Simva–TMZ-induced cell death is regulated via autophagy and UPR and connected to sterol regulatory element-binding proteins. Interestingly, chemoresistance in chordoma was shown to be related to PERK–autophagy crosstalk, and it was suggested that targeting this pathway could potentially overcome chemoresistance in these cells [71]. Seneca valley virus infection-induced autophagy is connected to UPR via the PERK and ATF6 arms, which shows the connection of autophagy to UPR in different cellular stress responses [71]. ATG proteins (ATG5, ATG7, and ATG5/ATG7 together) are involved in the regulation of autophagy, ER stress, and apoptosis through PERK signaling in chondrocytes; PERK has emerged to act as a hub to connect these three vital signaling pathways [72]. 

Based on our results, the Simva–TMZ combination decreases cell viability and the p-eIF2α/eIF2α ratio, strongly suggesting inhibition of the PERK arm. Co-treatment with PERKi could not change this potential effect. This suggests that the cell death mechanism of Simva–TMZ might be independent of inhibition of the PERK kinase activity at this dose and time point. Salubrinal co-treated with Simva-TMZ in GBM cells increased the ratio of p-eIF2α/eIF2α, as is to be anticipated with a phosphatase inhibitor, without any effects on the cell viability. It appears that although Simva–TMZ affects the e-IF2α phosphatase activity, this pathway is not involved in Simva–TMZ-induced cell death. 

U87 and U251 cells are different cell lines from point of view of p53 status. U87 is p53 wild type, and U251 is p53 mutant. They were both sensitive to TMZ treatment [73,74]. Based on several previous investigations, glioma cell lines that did not express a functional p53 were more sensitive to TMZ treatment [75,76,77]. Our apoptotic flow cytometry results indicate that TMZ induces cell death in U87 and U251 (7.74% and 21.5%, respectively), which could relate to p53 status in these cells. 

P53 regulates UPR and autophagy pathway-related proteins [26,78]. In cancer, a mutated p53 decreases autophagy and favors the proliferation of tumor cells. Accumulating evidence shows that p53 plays a dual role in the control of autophagy [79]. On one hand, nuclear p53 can induce autophagy by trans-activating autophagy-inducing genes. On the other hand, cytoplasmic p53 may act as a repressor of autophagy [79]. Therefore, the difference between U251 and U87 in autophagy flux inhibition via Simva and Simva–TMZ could be correlated to the difference of p53 between these two cell lines. 

Based on several recent studies, p53 mutant cancer cells have higher levels of IRE1, and the activation of XBP1 was induced in the absence of stress and activation and contributes to higher malignancy and the aggressive phenotype of the tumors [80]. It addition, it has been reported that p53 is involved in the regulation of cellular homeostasis during the UPR [81]. Therefore, a difference in UPR between U87 and U251 in the presence of Simva–TMZ treatment could be correlated to the p53 status in these cells. 

As shown in the Figure 1, Simva sensitizes GBM cells to TMZ-induced cell death via complex mechanisms, involving both autophagy and UPR. Co-treatment increases the GRP78, IRE-1, XBP-1s, and cleaved caspase-3 protein amount, as well as the protein amount of Beclin-1, p62, and LC3β-II, and it decreases the p-eIF2α/eIF2α ratio. Our findings show that although Simva–TMZ activates IRE1 RNase, inhibition of this arm does not have any effect on Simva–TMZ-induced cell death. On the other hand, Simva–TMZ-induced cell death changes the activity of the PERK–UPR arm in both U87 and U251 cells, but the inhibition of PERK only affects Simva–TMZ-induced cell death in U87 cells. Unequivocally, our results demonstrate the existence of an intricate connection involving IRE1 and PERK between UPR and autophagy in our in vitro GBM models. Above all, our current investigation highlights the potential for the use of statins in combination with a chemotherapeutic agent in the treatment of GBM. The current investigation showed that Simva–TMZ combination therapy induced UPR response in GBM cells, but it is not involved in its cytotoxic and apoptotic effects on GBM cells. Interestingly, our results showed that UPR inhibition inhibited the effect of Simva–TMZ on autophagy flux inhibition, but it did not change Simva–TMZ-induced cell death and apoptosis in GBM cells. It shows that Simva sensitizes GBM cells to TMZ-induced apoptosis via a complex mechanism beyond simple apoptosis, autophagy, and UPR crosstalk together and potentially depends on some unknown Simva pleiotropic effect and needs further investigations. Currently, our team has done the in vivo flank model of GBM (using U251 cells in immune-compromised mice) and was able to show that Simva significantly potentiates the anti-tumor effect of TMZ in a flank model. The application of statin as potential adjuvant chemotherapy needs critical in vitro, in vivo, and clinical investigation in the future, and we need further clinical trials to address this issue. 

## 4. Materials and Methods

### 4.1. Reagents and Drugs

Temozolomide (CAS Number: 85622-93-1), simvastatin (CAS Number: 79902-63-9), mevalonate (CAS Number: 674-26-0), propidium iodide (CAS Number: 25535-16-4), 3-(4,5-dimethyl-2-thiazolyl)-2,5-diphenyl-2H-tetrazolium bromide) (MTT) (CAS Number: 298-93-1), PERKi (PERK kinase activity inhibitor, GSK2606414, CAS 1337531-89-1), and salubrinal (phosphor-eIF2α phosphatase inhibitor, CAS 304475-63-6) were purchased from Sigma-Aldrich Co. (Oakville, ON, Canada). MKC8866 (IRE1 RNase inhibitor) was provided by MannKind company (Los Angeles, CA, USA). 

Rabbit and mouse primary antibodies against human, such as p-H2A.X, GRP78, IRE-1, ATF-6, XBP-1s, e-IF2α, P-e-IF2α, caspase-3, Beclin-1, LC3β-II, and p62 were purchased from Cell Signaling Technology Co. (Beverly, MA, USA). GAPDH was obtained from Santa Cruz Biotechnology Inc. (Dallas, Texas, USA) and secondary antibodies, anti-rabbit IgG (whole molecule), and anti-mouse immunoglobulin G (IgG) (Fab specific), conjugated with a peroxidase enzyme, were purchased from Sigma-Aldrich (Oakville, ON, Canada).

The enhanced chemiluminescence (ECL) Western blot (high sensitivity) substrate kit (ab133406) was acquired from Abcam (Cambridge, MA, USA). RNA extraction reagent, BIOZOL (Zhejiang, China), the cDNA synthesis kit Thermo Fisher Scientific (Winnipeg, MB, Canada), and SYBR Green QPCR master mix were purchased from Applied Biosystems (Waltman, MA, USA). The bicinchonic acid (BCA) protein assay kit was obtained Thermo Fisher Scientific (Winnipeg, MB, Canada).

### 4.2. Cell Lines, Culture, and Treatment

We obtained the human glioblastoma cell lines, U87 (p53 wild type) and U251 (p53 mutant), from the Bonyakhteh Company (Bonyakhteh, Tehran, Iran). We used cells in passages 3–6 for all experiments and bought fresh and authenticated cell lines from the company after 6 passages. The cells were cultured in Dulbecco’s Modified Eagle’s Medium high-glucose, high glutamine (DMEM) (Bio Idea, Tehran, Iran), supplemented with 10% Fetal Bovine Serum (FBS) (Gibco™; Cat #: 16000044) and 1% penicillin–streptomycin (Gibco, (Waltman, MA, USA), and maintained in a humidified incubator under 95% air and 5% CO_2_ at 37 °C.

### 4.3. MTT Assay

MTT assay was used to measure the viability of cells under different experimental conditions. The effect of Simva (0–20 μM), TMZ (0–1000 μM), and mevalonate (0–500 mM) on GBM viability were evaluated at different time points (24–96 h). The effect of MKC8866 (0–80 μM), PERKi (0–20 μM), and salubrinal (0–20 μM) on GBM on cell viability were assessed at 48 and 72 h. After determining the optimum treatment dose for mevalonate (2.5 mM), GSK-PERK (5 μM), salubrinal (15 μM), and MKCC8866 (30 μM), we evaluated the Simva-mediated sensitization of GBM cells to TMZ-induced cell death and the role of UPR herein. U251 and U87 cells were seeded in 96-well plates (5000 cells per well) and treated after reaching 40% confluency. We pretreated U251 and U87 cells with Simva (1 and 2.5 μm, respectively) for 4 h and then co-treated them with TMZ (100 µM) for 72 h. To assess the role of mevalonate, cells were pretreated with mevalonate (2.5 mM) for 4 h before treatment with Simva and then co-treated with TMZ for 72 h. MTT assays were performed as described previously. Briefly, we added 20 µL MTT to each well at the predetermined time points. We removed all the medium after 3 h and added 200 µL DMSO. Finally, after 20 min incubation, the absorbance was read at 570 nm [37,82]

### 4.4. Evaluation of Cell Death by Flow Cytometry

Cell death was evaluated using the Nicoletti method [83,84]. Briefly, U251 and U87 cells were cultured in 6-well plates in the presence of the least toxic dose of Simva (1 and 2.5 µM, respectively, based on MTT assay results). After incubating cells with Simva for 4 h, 100 µM TMZ was added (without changing the media), and cells were kept under these treatment conditions (Simva–TMZ) for 72 h. We will call this the procedure for our experimental protocol (OEP). To assess the effect of mevalonate, GBM cells were pretreated with 2.5 µM mevalonate (MEV) for 3 h before OEP. After 72 h, the cells were detached, and pellets were re-suspended in a hypotonic PI lysis buffer (0.1% Triton X-100, 1% sodium citrate, 0.5 mg/mL RNase A and 40 μg/mL propidium iodide) and incubated for 30 min at 37 °C. The apoptotic dead nuclei were defined by flow cytometry as a sub-G1 population. All calculations were made in 10,000-event count. 

### 4.5. Immunoblotting 

The protein assay, sample preparation, and SDS-PAGE electrophoresis were performed as described in our previous investigations [41,85,86,87]. Briefly, the cells were harvested and lysed with NP-40 lysis buffer. Based on the type of protein, 10–30 µg of total protein was subjected to electrophoresis on SDS-PAGE gels (15% for MW 60 KD and lower, and 10% for MW > 60 KD), and separated proteins were transferred to 0.2 μm nitrocellulose membranes (Bio-Rad; #1620112). After overnight blocking (5% fat-free milk), the membranes were incubated with primary antibodies (P-H2A (Ser 139), GRP78, IRE-1, ATF-6, XBP-1s, e-IF2α, p-eIF2α (Ser 51), caspase-3, Beclin-1, LC3β-II, p62, and GAPDH) overnight at 4 °C. Antibodies were used in dilutions according to the manufacturer’s protocol. After incubation with suitable secondary antibodies for 90 min at room temperature, the membranes were incubated with enhanced chemiluminescence (ECL) reagents (Abcam, Cambridge, MA, USA) and developed by the ChemiDoc^TM^ MP imaging system (Bio-Rad, Hercules, CA, USA). The intensity of blots was measured by Image Lab densitometry software, and all bands were normalized to the GAPDH protein amount to correct for marginal deviations in protein loading.

### 4.6. Inhibition of Specific UPR Signaling Arms (PERK & IRE1) 

We used MKC8866, a PERKi, and salubrinal to investigate the role of UPR in the regulation of Simva–TMZ-induced cell death in GBM cells. To determine the effective doses of the inhibitors, we treated GBM cells with 1–20 μM GSK–PERK, 1–20 μM salubrinal, and 10–80 μM MKC8866 for 72 h, and evaluated spliced XBP-1 (XBP-1s), e-IF2α, p-eIF2α by immunoblotting. After identifying inhibitor concentration with close to the maximal effects of each inhibitor, we used these concentrations to assess the role of UPR in the regulation of Simva–TMZ-induced cell death. 

### 4.7. Quantitative Real-Time PCR

Total RNA was isolated from 1 × 10^6^ cells (U87 and U251) by the BIOZOL, and RNA was converted to cDNA by the cDNA synthesis kit based on the manufacturer’s instructions. Quantitative real-time PCR was carried out using the SYBR Green PCR Master Mix with the following set of primers: GAPDH (as an internal control) forward, 5′-CGACCACTTTGTCAAGCTCA-3′, reverse, 5′-AGGGGTCTACATGGCAACTG-3′, and XBP-1s forward, 5′-TGCTGAGTCCGCAGCAGGTG-3′ and reverse, 5′-GCTGGCAGGCTCTGGGGAAG-3′. The data were analyzed by 7500 Software v 2.0.1. The relative expression level of the XBP-1s gene was calculated by the 2^−△△Ct^ formula. 

### 4.8. Statistical Analysis

Descriptive statistics, including mean ± SD, were calculated first. Inferential statistics (i.e., one-or two-way Analysis of Variance (ANOVA)) were determined thereafter. Finally, a Bonferroni’s post hoc test was used to detect the statistical significance of differences (*p*-value < 0.05) by GraphPad Prism software v. 6.0. All experiments were performed in at least 3 different biological replicates.

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
