# Peer review of "Simvastatin Induces Unfolded Protein Response and Enhances Temozolomide-Induced Cell Death in Glioblastoma Cells"

_cells, 2020, doi:10.3390/cells9112339_

Round 1

Reviewer 1 Report

The role of the hydroxymethylglutaryl CoA inhibitor Simvastatin, in sensitizing human glioblastoma U87 and U251 cells to temozolomide (TMZ)-induced cell death has been reported in recent papers by the same research group. Simvastatin was found to be able of increasing TMZ-induced apoptosis of tumor cells by inhibiting autophagy and independently from the mevalonate cascade inhibition.

Considering the role of the Unfolded Protein Response (UPR) in autophagy regulation, the UPR involvement in Simva-TMZ-induced U87 and U251 cell death has been investigated in the present study, by using pharmacological inhibitors of IRE1α and PERK (UPR sensor proteins) or the eIF2α inhibitor salubrinal.

The experiments are quite well conducted, and the results appear interesting. However, the paper is very difficult to read in its current form. (i) In the introduction paragraph, the scientific hypothesis at the basis of the investigation as well as the experimental design must be more clearly defined. I would strongly suggest a “results and discussion” setting, also because (iii) the discussion is too long and largely consisting of introductory concepts. Moreover, the discussion is weak, missing actually, as far as the different response of the two GBM cell lines. (iv) A relevant part of supplementary material seems to be represented by already published results, and its location at the beginning of the results section further complicates the paper reading.

Author Response

The role of the hydroxymethylglutaryl CoA inhibitor Simvastatin, in sensitizing human glioblastoma U87 and U251 cells to temozolomide (TMZ)-induced cell death has been reported in recent papers by the same research group. Simvastatin was found to be able of increasing TMZ-induced apoptosis of tumor cells by inhibiting autophagy and independently from the mevalonate cascade inhibition.

Considering the role of the Unfolded Protein Response (UPR) in autophagy regulation, the UPR involvement in Simva-TMZ-induced U87 and U251 cell death has been investigated in the present study, by using pharmacological inhibitors of IRE1α and PERK (UPR sensor proteins) or the eIF2α inhibitor salubrinal.

The experiments are quite well conducted, and the results appear interesting. However, the paper is very difficult to read in its current form.

1) In the introduction paragraph, the scientific hypothesis at the basis of the investigation as well as the experimental design must be more clearly defined.

Answer: Authors appreciate your thoughtful comment, we provided our scientific hypothesis and a brief of our experimental design in lines 92-97.

2) I would strongly suggest a “results and discussion” setting, also because the discussion is too long and largely consisting of introductory concepts. Moreover, the discussion is weak, missing actually, as far as the different response of the two GBM cell lines (lines 412-428).

Answer: We appreciate the comment to improve the result section and summarize discussion while adding a paragraph about the differences between cell lines and difference in response to some of the treatments.

As it is important for the readers to provide information about the background of our treatment strategies, we combined sections 2.1, 2.2, and 2.3 in results into one section (2.1) in the revised article (lines 100-131). We also summarized the discussion and removed some parts which was focused on Simva and TMZ single treatment and make the discussion more focused on Simva-TMZ treatment and add a new paragraph focusing on the difference of these cell lines. We explained their difference on p53 [U87 (p53 wild-type), and U251 (p53 mutant)] and explained the how UPR and autophagy could potentially be regulated by p53 in these cells to discuss the difference of autophagy and UPR between two cell lines based on p53 status (lines 412-428). 

3) A relevant part of supplementary material seems to be represented by already published results, and its location at the beginning of the results section further complicates the paper reading

Answer: We appreciate the feedback. As it is important for the readers to provide information about the background of our treatment strategies, we combined sections 2.1, 2.2, and 2.3 in results into one section (2.1) in the revised article (lines 99-130).

Reviewer 2 Report

Cells (Manuscript ID: cells-936586), Comments to the Authors:

Title: Simvastatin Induces Unfolded Protein Response and Enhances Temozolomide-induced Cell Death in Glioblastoma Cells

Comments

The submitted manuscript investigated the involvement of UPR in Simva-TMZ-induced cell death by utilizing IRE1 RNase activity inhibitor MKC8866, PERK inhibitor GSK-2606414 (PERKi) and eIF2α inhibitor‎ salubrinal. Simva-TMZ treatment decreased viability of GBM cells and significantly increased apoptotic cell death when compared to TMZ or Simva alone. Simva-TMZ induced both UPR, as determined by an increase in GRP78, XBP splicing, eIF2αeIF2α phosphorylation, and inhibited autophagic flux (accumulation of LC3β-II and inhibition of p62 degradation). IRE1 RNase inhibition did not affect Simva-TMZ-induced cell death, but significantly induced p62 degradation and increased the LC3β-II/LC3β-I ratio in U87 cells while salubrinal did not affect Simva-TMZ induced cytotoxicity of GBM cells. In contrast, PERK inhibition significantly increased Simva-TMZ-induced cell death in U87 cells. Interestingly, whereas PERK inhibition induced p62 accumulation in both GBM cells lines, it differentially affected the LC3β-II/LC3β-I ratio in U87 (decrease) and U251 (increase) cells.

I think the submitted manuscript can be accepted for publication after the authors respond to the following comments:

  1. The authors mentioned the following phrase which is not clear “including macroautophagy (hereafter autophagy)”.
  2. The authors mentioned "we used 100 μM to keep TMZ concentration at a stable level during the period" What do you mean by a stable level in this context.
  3. Why did the authors choose to show the results in Fig 2 after 72 h. The authors should show the results after 24 and 48 h.
  4. In Fig. 3s, the authors showed that Simva was more active that TMZ in inducing apoptosis in U87 and U251. Can the authors comment on this observation?
  5. In Fig. 3s, why did the authors use Simva at a concentration of 2.5 micromole for U87 cells and 1 micromole for U251 cells.
  6. In Fig 4 s, the authors mentioned that "Simva-TMZ inhibit autophagic flux in GBM" can the authors comments on this observation in the light of their results on Beclin-1 and p62. Their discussion of “autophagic flux” is not clear throughout the manuscript.
  7. The following sentence is not clear and needs rephrasing, “Interestingly, co-treatment 129 significantly increased the P-eIF2α/eIF2α ratio compared to TMZ or Simva in U87 cells (Fig. 1, Fig. 5s 130 A-E), whereas this ratio decreased after co-treatment as compared to TMZ and increased in 131 comparisons with Simva in U251 cells”.
  8. The authors mentioned that the MTT assay results did not match the PI assay so they have to evaluate the cytotoxicity using other types of cell viability assay.
  9. The authors should discuss the potential application of Simva-TMZ combination. The use of statins as anticancer did not show remarkable results and their effects were non-significant in trials.
  10. There are many typos and grammatical mistakes in the manuscript that should be corrected before resubmission.

Author Response

Reviewer #2

The submitted manuscript investigated the involvement of UPR in Simva-TMZ-induced cell death by utilizing IRE1 RNase activity inhibitor MKC8866, PERK inhibitor GSK-2606414 (PERKi) and eIF2α inhibitor‎ salubrinal. Simva-TMZ treatment decreased viability of GBM cells and significantly increased apoptotic cell death when compared to TMZ or Simva alone. Simva-TMZ induced both UPR, as determined by an increase in GRP78, XBP splicing, eIF2αeIF2α phosphorylation, and inhibited autophagic flux (accumulation of LC3β-II and inhibition of p62 degradation). IRE1 RNase inhibition did not affect Simva-TMZ-induced cell death, but significantly induced p62 degradation and increased the LC3β-II/LC3β-I ratio in U87 cells while salubrinal did not affect Simva-TMZ induced cytotoxicity of GBM cells. In contrast, PERK inhibition significantly increased Simva-TMZ-induced cell death in U87 cells. Interestingly, whereas PERK inhibition induced p62 accumulation in both GBM cells lines, it differentially affected the LC3β-II/LC3β-I ratio in U87 (decrease) and U251 (increase) cells.

I think the submitted manuscript can be accepted for publication after the authors respond to the following comments:

  • The authors mentioned the following phrase which is not clear “including macroautophagy (hereafter autophagy)”.

Answer: We appreciate the close attention of the respected reviewer. As we have three types of autophagy including macroautophagy, microautophagy, and chaperon mediated autophagy (CMA), we identified the type of autophagy that we investigated in the beginning of our manuscript to make it clear for the readers of our manuscript.

  • The authors mentioned "we used 100 μM to keep TMZ concentration at a stable level during the period" What do you mean by a stable level in this context.

Answer: It was an error in our primary submission and we have deleted the statement in the revision. We have explained the rational behind choosing specific concentration of Simva, and TMZ in lines 106-108 of the revised manuscript.

  • Why did the authors choose to show the results in Fig 2 after 72 h. The authors should show the results after 24 and 48 h.

Answer: As we planned to address the potential effect of UPR arms in regulation of autophagy inhibition by Simva-TMZ treatment, we chose 72 h to be consistent with our previous investigation on this topic that we have addressed Simva-TMZ-indcued autophagy flux inhibition (Shojaei et al, 2020, FEBS J).

  • In Fig. 3s, the authors showed that Simva was more active that TMZ in inducing apoptosis in U87 and U251. Can the authors comment on this observation?

Answer: TMZ induced apoptosis is depended on DNA damage and its downstream response and translation which is usually a slow process while Simva-induced apoptosis is depended on the inhibition of small Rho GTPase protein which is a fast-cellular event. It is why we observe higher apoptosis induction compared to TMZ in 72 h time point. Our unpublished data showed that TMZ and Simva induced apoptosis is almost the same in longer time points. We added a sentence in the revised manuscript to explain this observation (lines 118-120).   

  • In Fig. 3s, why did the authors use Simva at a concentration of 2.5 micromole for U87 cells and 1 micromole for U251 cells?

Answer: We addressed this important question in the revised manuscript. “We selected Simva, and TMZ concentrations in each cell line based on their cell viability in 72 h (70% < cell viability < 95%) so the single treatment has toxicity less than 30%.” (Lines 106-108).

  • In Fig 4 s, the authors mentioned that "Simva-TMZ inhibit autophagic flux in GBM" can the authors comments on this observation in the light of their results on Beclin-1 and p62. Their discussion of “autophagic flux” is not clear throughout the manuscript.

Answer: Autophagy flux refers to delivery of autophagosome to lysosomes for degradation. Based on the “Autophagy Guideline” (Klionsky et al, 2008, 2012, 2016, 2020, Autophagy), it is measured via amount of LC3β-II and p62 in immunoblotting. Beclin-1 is not considered a suitable marker for autophagy flux measurement. We added a sentence in the revised manuscript to explain it for the readers (lines 128-130).

  • The following sentence is not clear and needs rephrasing, “Interestingly, co-treatment 129 significantly increased the P-eIF2α/eIF2α ratio compared to TMZ or Simva in U87 cells (Fig. 1, Fig. 5s 130 A-E), whereas this ratio decreased after co-treatment as compared to TMZ and increased in 131 comparisons with Simva in U251 cells”.

Answer: We rephrased the sentence to “Simva-TMZ co-treatment significantly increased p-eIF2α/eIF2α ratio in U87 while decreased it in U251 compared to Simva and TMZ single treatment (Fig. 1, Fig. 5s A-E, Fig. 1, and Fig. 6s A-E) (lines 137-139).

  • The authors mentioned that the MTT assay results did not match the PI assay so they have to evaluate the cytotoxicity using other types of cell viability assay.

Answer: We appreciate the respected reviewer comments. We have confirmed our MTT results with apoptosis assay in all treatment and added new data (Fig. 2E-F, Fig. 5 E-F, Fig. 8 E-F).

  • The authors should discuss the potential application of Simva-TMZ combination. The use of statins as anticancer did not show remarkable results and their effects were non-significant in trials.

Answer: We appreciate the respected reviewer feedback and added a paragraph at the end of the discussion to highlight future directions in this topic (lines 441-449).

  • There are many typos and grammatical mistakes in the manuscript that should be corrected before resubmission.

Answer: Thank you for your constructive comment, we corrected all typos and grammatical mistakes in the manuscript in red font.

Reviewer 3 Report

Manuscript ID: cells-936586

Title: Simvastatin Induces Unfolded Protein Response and Enhances Temozolomide-induced Cell Death in Glioblastoma Cells

Authors: Sanaz Dastghaib , Shahla Shojaei , Zohreh Mostavavi-Pour , Pawan Sharma , John B Patterson , Afshin Samali , Pooneh Mokarram , Saeid Ghavami

The manuscript Dastghaib et al. investigates in more details the signalling pathway involved in previously reported simvastatin (Simva) increased TMZ-induced apoptosis via inhibition of autophagic flux in GBM cells. The authors investigated the role of the unfolded protein response (UPR) pathway in the regulation of autophagy, the involvement of UPR in Simva-TMZ-induced cell death by utilizing highly selective IRE1 RNase activity inhibitor MKC8866, PERK inhibitor GSK-2606414 (PERKi) and eIF2α inhibitor‎ salubrinal. They showed that PERK inhibition significantly increased Simva-TMZ-induced cell death in U87 cells. It is interesting that whereas PERK inhibition induced p62 accumulation in both GBM cells lines, it differentially affected the LC3β-II/LC3β-I ratio in U87 (decrease) and U251 (increase) cells. Simvastatin sensitizes GBM cells to TMZ-induced cell death via a mechanism that involves autophagy and UPR pathways. Their results imply that the IRE1 and PERK signalling arms of the UPR regulate Simva-TMZ-mediated autophagy flux inhibition in U251 and U87 GBM cells.

I find the manuscript interesting and valuable to be published in Cells. There are some questions which need to be addressed as well as several minor issues which need to be corrected.

Major:

  • The fact that authors described two glioblastoma cells as either p53 wild type or p53 mutant, I am wondering why they do not discuss this specific status in correlation with obtained data especially because they do not show similar pattern of cell response towards PERK inhibition.

Minor:

  • Line 98: correct Simvainin in Simvastatin
  • Line 101: correct 100μL in 100 μL
  • Line 130: correct P-eIF2α/eIF2α in p-eIF2α/eIF2α (similar mistakes are performed in Figure 1 (western blot for p-eIF2α); P62 correct in p62 (Figure 3A,D,G; Figure 7 AB, D; Figure 9A; Line 405, Line 409, Line 492, Line 503; Scheme P62 correct in p62);
  • The many times repeated label …Simva 1 or 2.5 μM (Figure 3A, Figure 4F,G,H etc.) is not the best way to label the use of different concentrations on Simva for different cell types especially since in Material and Methods part it is written that U251 was treated with 1 and U87 with 2.5 μM. Very often from the figures it can be read opposite. Authors need to find better way to label the data.
  • In general all Figures are prepared pretty clumsy. They are not matching in size, the labels are written clumsy (lot of spaces are missing between numbers and μM; or numbers and h (Figure 6B,C; Figure 9B,C)
  • Line 215: correct 5μM in 5 μM
  • Correct Figure 5A-D: Perk In in PERKi; Figure 6A-C
  • Sometimes the authors are using h or hrs. Please decide which label for time you would like to use.
  • Line 264: correct 48h in 48 h
  • Line 268: correct p-eIF2α/ eIF2α in p-eIF2α/eIF2α
  • Line 274: 2.5 μM is missing
  • Line 373: correct Hela in HeLa
  • Correct through whole text XY°C in XY °C
  • Scheme: correct Reticulum Endoplasmic in Endoplasmic Reticulum
  • The font is not uniform through the text

Author Response

Reviewer #3:

The manuscript Dastghaib et al. investigates in more details the signalling pathway involved in previously reported simvastatin (Simva) increased TMZ-induced apoptosis via inhibition of autophagic flux in GBM cells. The authors investigated the role of the unfolded protein response (UPR) pathway in the regulation of autophagy, the involvement of UPR in Simva-TMZ-induced cell death by utilizing highly selective IRE1 RNase activity inhibitor MKC8866, PERK inhibitor GSK-2606414 (PERKi) and eIF2α inhibitor‎ salubrinal. They showed that PERK inhibition significantly increased Simva-TMZ-induced cell death in U87 cells. It is interesting that whereas PERK inhibition induced p62 accumulation in both GBM cells lines, it differentially affected the LC3β-II/LC3β-I ratio in U87 (decrease) and U251 (increase) cells. Simvastatin sensitizes GBM cells to TMZ-induced cell death via a mechanism that involves autophagy and UPR pathways. Their results imply that the IRE1 and PERK signalling arms of the UPR regulate Simva-TMZ-mediated autophagy flux inhibition in U251 and U87 GBM cells.

I find the manuscript interesting and valuable to be published in Cells. There are some questions which need to be addressed as well as several minor issues which need to be corrected.

Major:

-The fact that authors described two glioblastoma cells as either p53 wild type or p53 mutant, I am wondering why they do not discuss this specific status in correlation with obtained data especially because they do not show similar pattern of cell response towards PERK inhibition.

Answer: This is a very important comment. We have added a section to the discussion to highlight the difference of these two-cell line based on p53 and its possible effect on our observation in regulation of UPR and autophagy (lines 412-428).

Minor:

-Line 98: correct Simvainin in Simvastatin

Answer: Thank you for reminding this point; we have corrected it (line 99)

-Line 101: correct 100μL in 100 μL

Answer: Thank you for reminding this point; we have corrected it (line 103)

-Line 130: correct P-eIF2α/eIF2α in p-eIF2α/eIF2α (similar mistakes are performed in Figure 1 (western blot for p-eIF2α); P62 correct in p62 (Figure 3A,D,G; Figure 7 AB, D; Figure 9A; Line 405, The many times repeated label …Simva 1 or 2.5 μM (Figure 3A, Figure 4F,G,H etc.) is not the best way to label the use of different concentrations on Simva for different cell types especially since in Material and Methods part it is written that U251 was treated with 1 and U87 with 2.5 μM. Very often from the figures it can be read opposite. Authors need to find better way to label the data.

In general, all Figures are prepared pretty clumsy. They are not matching in size, the labels are written clumsy (lot of spaces are missing between numbers and μM; or numbers and h (Figure 6B, C; Figure 9 B, C).

Answer: We have corrected all figures (main figure and supplementary figures) based on the respected reviewer comment and deleted the concentrations as they have been mentioned in the M&M and figure legends. We used new figures in our manuscript.

-Line 409, Line 492, Line 503; Scheme P62 correct in p62);

Answer: Thank you for reminding this point; we have corrected p62 through whole text and Scheme.

-Line 215: correct 5μM in 5 μM

Answer: Thank you for reminding this point; we have changed (line 244)

-Correct Figure 5A-D: Perk In in PERKi; Figure 6A-C

Answer: Thank you for reminding this point, we have corrected it (Figure 5, 6).

Sometimes the authors are using h or hrs. Please decide which label for time you would like to use.

-Line 264: correct 48h in 48 h

Answer: Thank you for reminding this point; we have changed (line 296)

-Line 268: correct p-eIF2α/ eIF2α in p-eIF2α/eIF2α

Answer: Thank you for reminding this point; we have changed (line 310)

-Line 274: 2.5 μM is missing

Answer: Thank you for reminding this point, we have corrected to “1 µM in U87 cells in 48 h”. (line 286) 

-Line 373: correct Hela in HeLa

Answer: Thank you for reminding this point; we have changed it (line 373)

-Correct through whole text XY°C in XY °C

Answer: Thank you for reminding this point; we have changed it through whole text (line 507,517)

-Scheme: correct Reticulum Endoplasmic in Endoplasmic Reticulum

Answer: Thank you for reminding this point, we have corrected inside Scheme

-The font is not uniform through the text

Answer: Thank you for reminding this point, we have fixed it through text.

Round 2

Reviewer 1 Report

The quality of the paper has been signifivantly improved

Reviewer 2 Report

Cells (Manuscript ID: cells-936586), Comments to the Authors:

Title: Simvastatin Induces Unfolded Protein Response and Enhances Temozolomide-induced Cell Death in Glioblastoma Cells

Comments

After reading the authors' response to my comments, I think the manuscript can be accepted for publication.

This manuscript is a resubmission of an earlier submission. The following is a list of the peer review reports and author responses from that submission.

Round 1

Reviewer 1 Report

In this manuscript the authors build a case for simvistatin an commonly clinically used HMG Co A reductase inhibitor for high cholesterol as a synergistic therapeutic with standard of care temozolomide. The mechanism builds on previous in vitro work where they found that simvistatin sensitizes glioblastoma cell lines to temozolomide. They systematically interrogate each of the pathways of the UPR and their mediators. Using cell viability assays and inhibitors of each arm. This approach is methodical but would be improved by a transient genetic knockdown approach to confirm specificity (siRNA). The study is performed on 2 widely used ATCC cell lines U87 and U251. It could improve relevance by adding validated primary cell lines as well. All work was done in vitro and would benefit from a set of experiments done in vivo (such as a immunodeficient xenografting model). 

In Figure 1 the cleaved caspase for U251 Lane for Simi 1uM has more expression than temozolomide alone. In U87 there is similar expression in all lanes. Could you please explain these results? 

Author Response

Reviewer #1

Reviewer #1

C1. In this manuscript, the authors build a case for simvistatin an commonly clinically used HMG Co A reductase inhibitor for high cholesterol as a synergistic therapeutic with standard of care temozolomide. The mechanism builds on previous in vitro work where they found that simvistatin sensitizes glioblastoma cell lines to temozolomide. They systematically interrogate each of the pathways of the UPR and their mediators. Using cell viability assays and inhibitors of each arm. This approach is methodical but would be improved by a transient genetic knockdown approach to confirm specificity (siRNA). The study is performed on 2 widely used ATCC cell lines U87 and U251. It could improve relevance by adding validated primary cell lines as well. All work was done in vitro and would benefit from a set of experiments done in vivo (such as a immunodeficient xenografting model).

R1. We appreciate these valuable comments. A) In our recent manuscript (Shojaei et al, 2020, FEBS J) we have used human primary GBM cells to confirm findings in U87 and U251 cells. All findings in the primary cells were in line with U87 and U251 cells including autophagy flux inhibition and sensitizing GBM cells to TMZ-induced apoptosis. In the current study, we have not used human primary GBM cells as our results in U87 and U251 were confirmed in primary GBM cells in the previous study. We also acknowledge reviewer’s comment on using siRNA to confirm our inhibitors’ results. It is a good suggestion, we believe we used a range of different concentrations to avoid off-target effect of the inhibitors in this study. Additionally were were highly interested in using MKC8866 (a specific inhibitor of IRE1 RNase activity) and evaluate its effect on RNase function of IRE1 in Simva-TMZ treatment therefore we have not used siRNA in this study. We have also done flank model of U251 in mouse immunodeficient model and measure autophagy/UPR and measured lipidomic of the tumors. Right now, we are preparing the manuscript for that study and it will be submitted soon.

C2. In Figure 1 the cleaved caspase for U251 Lane for Simi 1uM has more expression than temozolomide alone. In U87 there is similar expression in all lanes. Could you please explain these results?

R2. We thank reviewer to clarify this. Many investigators recently used Cas-3/-7 combination assay (Caspase-Glo promega assay) to determine apoptosis in U87 and U251 cells (Shojaei etal FEBS J, 2020; Thanasupawat et al, Mol Oncol, 2018; Ma et al, Ann Clin Lab Sci, 2020; and Viswanathan et al, Eur J Pharmacol, 2020). In previous investigations it has been showed that Caspase-7 is more active than Caspase-3 in U87 in response to apoptosis stimuli (Keing et al, Apoptosis, 2011). In our current study we used cleaved caspase-3 and showed that combination of Casp-3/-7 measurement in U87 gave the same amount of Casp-3/-7 activation in both U87 and U251 cells (Shojaei et al, FEBS J, 2020, Figure 8 A and B). Therefore, we believe that Caspase-7 activation could be a key determining step in overall Cas-3/-7 activity in response to apoptosis.

Reviewer 2 Report

MANUSCRIPT REVIEW

In the study entitled “Simvastatin Induces Unfolded Protein Response and Enhances Temozolomide-induced Cell Death in Glioblastoma Cells”, the authors investigated the involvement of UPR in Simva-TMZ-induced cell death by utilizing highly selective IRE1 RNase activity inhibitor MKC8866, PERK inhibitor GSK-2606414 (PERKi) and eIF2α inhibitor salubrinal. Concluding, tha authors  affirmed that Simvastatin sensitizes GBM cells to TMZ-induced cell death via a mechanism that involves autophagy and UPR pathways. More specifically, our results imply that the IRE1 and PERK signaling arms of the UPR regulate Simva-TMZ-mediated autophagy flux inhibition in U251 and U87 GBM cells.

The work is interesting and some aspects are innovative. The research project and the experimental design are appropriate, the conceptional structure is well organized. However, I recommend major revision because the introduction should be better developed, figures are poor quality and “Materials and methods” section shows several lacks.

Point by point

INTRODUCTION

In general, the introduction presents a poor detailed background. About my opinion, you should better describe the current findings: the references cited in this section are appropriate but you should develop the speech.

FIGURES

In general, the figures are low quality and it’s difficult to read the writings. Also, there are too many figures: you could put together some of pictures.

Figure 1. You should add the densitometric analysis.

MATERIALS AND METHODS

In general, sometimes you reported the companies of products and/or cities sometimes not. Please, make it uniform (e.g. company/city/state).

Line 449. In the “Cell Lines, Culture and Treatment” subsection, you should indicate if, when and how cell line was last authenticated, maximum number of passages before the cells were analyzed.

Line 474. How many events did you acquire?

Line 479. What is the SDS-PAGE percentage?

Line 480. What kind of blocking solution (e.g.milk, albumine)?

Author Response

Reviewer #2

In the study entitled “Simvastatin Induces Unfolded Protein Response and Enhances Temozolomide-induced Cell Death in Glioblastoma Cells”, the authors investigated the involvement of UPR in Simva-TMZ-induced cell death by utilizing highly selective IRE1 RNase activity inhibitor MKC8866, PERK inhibitor GSK-2606414 (PERKi) and eIF2α inhibitor salubrinal. Concluding, tha authors  affirmed that Simvastatin sensitizes GBM cells to TMZ-induced cell death via a mechanism that involves autophagy and UPR pathways. More specifically, our results imply that the IRE1 and PERK signaling arms of the UPR regulate Simva-TMZ-mediated autophagy flux inhibition in U251 and U87 GBM cells. The work is interesting and some aspects are innovative. The research project and the experimental design are appropriate, the conceptional structure is well organized. However, I recommend major revision because the introduction should be better developed, figures are poor quality and “Materials and methods” section shows several lacks.

Point by point comments

INTRODUCTION

C1. In general, the introduction presents a poor detailed background. About my opinion, you should better describe the current findings: the references cited in this section are appropriate but you should develop the speech.

R1: In the revised version we have  developed the tone of introduction and made it simpler and informative. These changes are highlighted in red in the revised version of manuscript.

FIGURES

C2. In general, the figures are low quality and it’s difficult to read the writings. Also, there are too many figures: you could put together some of pictures.

Answer: We apologize for the inconvenience due to reduction in resolution after being converted to PDF. We have now prepared these figures with higher resolution and larger fonts. All new figures are included in the revised manuscript.

C3. Figure 1. You should add the densitometry analysis.

R3: We have included the densitometry analysis for Figure 1 in supplementary Figure 5s and 6s in the revised manuscript.

MATERIALS AND METHODS

C4. In general, sometimes you reported the companies of products and/or cities sometimes not. Please, make it uniform (e.g. company/city/state).

R4: We have corrected the requested information regarding M&M in the revised manuscript (red font, section 4.1).

C5. Line 449. In the “Cell Lines, Culture and Treatment” subsection, you should indicate if, when and how cell line was last authenticated, maximum number of passages before the cells were analyzed.

R5: We have added the information about the passages and cell line authentication to the M&M (red font, line 464-465).

C6. Line 474. How many events did you acquire?

R6: We have added a sentence indicating the event count (10,000) to the M&M (red font, line 495).

C7. Line 479. What is the SDS-PAGE percentage?

R7: We have added information about the percentage of SDS-PAGE gel (10 and 15%) to the M&M (red font, line 500).

C8. Line 480. What kind of blocking solution (e.g.milk, albumine)?

R8: We have added blocking condition (5% fat free milk) to the M&M (red font, line 502).

Reviewer 3 Report

Dastghaib et al studied in a p53 wildtype and a p53-mutated human glioblastoma cell line the temozolomide-sensitizing effect of the statin vimvastatin and the role of autophagy unfolded protein response herein. Methodologically, cell viability was assessed by MTT assay (and in one experiment by subG1 fraction of in Nicoletti staining in flow cytometry), underlying mechanisms by rescuing cholesterin biosynthesis with mevalonate, inhibiting IRE-1alpha with MKC8866, PERK with GSK2606414, and p-eIF2alpha with salubrinal. Cell viability experiments were complemented by immunoblotting of the involved proteins ant RT-PCR analysis of XPB-1s frame-shift variant. The authors conclude that simvastatin sensitizes glioblastoma cells to temozolomide via pathways involved in autophagy and unfolded protein response.

This is a well-written and meticulously performed study with mostly comprehensible data presentation and statistical analysis. I have three major concerns: 1. The epidemiological evidence for a better survival of glioblastoma patients with statin prescription is poor which lowers the clinical relevance of the data. 2. MTT assay is quick but the interpretation of the MTT data is limited. From an oncological point of view, clonogenic survival is a much better endpoint to evaluate treatment efficacy since it determines whether or not a tumor may relapse after end of therapy. Therefore, I strongly recommend that the authors confirm their key findings by determining clonogenic survival. 3. Provided that the tumor-eradicating efficacy of the combination therapy can somehow be deduced from the MTT data, the additional (additive) effect of statins (about 15% less viable cells as compared to temozolomide treatment alone) is not huge questioning a strategy to combine statins with temozolomide.

Specific comments

Introduction

-Page 1, lines 60-61: “Epidemiological studies have shown statins reduce mortality rates of different types of cancer, including GBM [15,16].” Epidemological evidence for a statin use-mediated survival benefit is debated (J Clin Neurosci. 2017, 42:196-197). The reference #16 (in contrast to the statement of the authors, see Suppl. Figure S14 of this reference) and a more recent study (J Clin Neurosci, 31 (2016), pp. 176-180) do not show a survival benefit for brain tumor and glioblastoma patients, respectively, with statin prescription. Please, provide more detailed information.

-Page 2, line 52: “..with a median survival rate of less than 3% for 12−15 months [7-9].” This statement is not correct

-Page 2, lines 55, 56: “However, the survival chances of GBM  patients remained low (~21 months) because of development of resistance to TMZ treatment 56 [9,13]” 21 months??? See sentence above. Please, introduce the temozolomide resistance gene MGMT and the MGMT promoter methylation in a subgroup of glioblastoma patients. By the way, standard therapy according to Stupp et al. comprises surgery, temozolomide radiochemotherapy and temozolomide maintenance chemotherapy.

Results

-Page 2, line 93:Peak simvastatin plasma concentration (after oral dose of 40 mg Br J Clin Pharmacol. 2009,67:520–526) is in the range of 30 nM. Why do the authors apply micromolar concentrations?

Figure 1:

-The representative plots given are not very convincing. I cannot see the effects evident from the semiq-uantification of the bands (Figs. 5s and 6s). The authors do not describe and discuss the differences between the 0 h and 72 h control. Moreover, some plots are too dark. Some plots give a single band for U87 cells and a double band for U251. Please exchange the plots by more representative ones (Figs 5s,C and 6s,C: XBP-1s instead of XBP-1)

Figure 2:

-This applies to all figures. Please, increase the fond size of the axis labeling.

-As discussed also by the authors in the discussion section, MTT assay measures NAD(P)H-dependent formation of formazan and is not the best surrogate marker for viability especially when interfering with autophagy. Why didn’t the author determine clonogenic survival by colony formation assay which is the gold standard in oncology?

-the applied temozolomide concentration above the clinically achievable concentration in the tumor. Nevertheless, the effect of temozolomide on formazan formation is only some 20% (also shown in Fig. 1s and 2s) although U251 and U87MG cells reportedly are MGMT negative (Chin J Cancer. 2014,33:115–122) Please, discuss.

-line 154 (and all further Legends). µ is mising

Figure 3s:

-subG1 population in Nicoletti staining is not specific for apoptotic cell death. This is also evident from the comparison of the subG1 population (Fig. 3s) and cleaved caspase-3 (Figs. 5s, F and 6s, F) data. Please reword (cell death instead of apoptosis), also throughout the main manuscript.

-Results, page 3, line 105: It is worth nothing

-Results page 3, line 111: ilipidated ? lipid-conjugated

-Results, page 6, lines 179ff and throughout the further text: “…we concluded that XBP-1 splicing is involved in the Simva-TMZ-mediated inhibition of autophagy flux, but that it does not affect the cytotoxic effects of Simva-TMZ”. , The “cytotoxic” effects are merely based on formazan formation which is a very week read-out (authors showed in Figs 2s and 3s discrepancies between formazan formation and DNA degradation). Please, confirm data with more conclusive endpoints such as colony formation assay.

-Figure 6: The pattern of the bands in figure part A is not consistent with the bar diagrams shown in B and C. E.g., the high abundances of p-eIF2alpha in U87 cells upon PERK inhibition + temozolomide  is not represented by the p-eIF2alpha/eIF2alpha ratio. The latter is similar to PERK inhibition + simvastatin which shows only a weak band of p-eIF2alpha in A. Please, give more representative blots.

Discussion

-page 14, line 334 (and throughout the text): “… IRE-1 expression” better protein amount or abundance

-page 14, line 336ff  “…Simva-TMZ elevated cleaved caspase-3 and p-H2A.X (an indication of double-strand 336 breaks) more than TMZ alone, especially in U251 cells. This implies that the effect of Simva-TMZ on ..” data on gammaH2AX are not described in the result section and don not seem very robust. In U87, simvastatin/temozolomide co-treatment reduces gammaH2AX abundance (Fig. 5s) and how do the authors explain the increase in gammaH2AX abundance just by 72 h cell culture?

Material and Methods

-Page 17, line 448: 5% CO2  - DMEM medium usually contains 3.4 g/l bicarbonate and requires higher CO2 concentrations to equilibrate pH 7.4.

Author Response

Reviewer #3

Dastghaib et al studied in a p53 wildtype and a p53-mutated human glioblastoma cell line the temozolomide-sensitizing effect of the statin vimvastatin and the role of autophagy unfolded protein response herein. Methodologically, cell viability was assessed by MTT assay (and in one experiment by subG1 fraction of in Nicoletti staining in flow cytometry), underlying mechanisms by rescuing cholesterin biosynthesis with mevalonate, inhibiting IRE-1alpha with MKC8866, PERK with GSK2606414, and p-eIF2alpha with salubrinal. Cell viability experiments were complemented by immunoblotting of the involved proteins ant RT-PCR analysis of XPB-1s frame-shift variant. The authors conclude that simvastatin sensitizes glioblastoma cells to temozolomide via pathways involved in autophagy and unfolded protein response. This is a well-written and meticulously performed study with mostly comprehensible data presentation and statistical analysis. I have three major concerns: 1. The epidemiological evidence for a better survival of glioblastoma patients with statin prescription is poor which lowers the clinical relevance of the data. 2. MTT assay is quick but the interpretation of the MTT data is limited. From an oncological point of view, clonogenic survival is a much better endpoint to evaluate treatment efficacy since it determines whether or not a tumor may relapse after end of therapy. Therefore, I strongly recommend that the authors confirm their key findings by determining clonogenic survival. 3. Provided that the tumor-eradicating efficacy of the combination therapy can somehow be deduced from the MTT data, the additional (additive) effect of statins (about 15% less viable cells as compared to temozolomide treatment alone) is not huge questioning a strategy to combine statins with temozolomide.

We appreciate these comments. Here is our specific answer to above comments:

C1. The epidemiological evidence for a better survival of glioblastoma patients with statin prescription is poor which lowers the clinical relevance of the data.

R1: There are several epidemiological studies regarding survival of the different cancer patients using statins including an important study which was published in NEJM in 2012 (PMID: 23134381). There is a direct study in 2013, which showed “Prediagnostic statin use was associated with a reduced HR of death (0.79; 95% CI: 0.63-1.00)” (PMID: 25455652). In a metanalysis it was also showed long term use of statin was a key factor for survival rate in GBM patient (PMID: 32118710) although it was not significant. Based on our experience the time that patients use statin is a key factor to determine the improvement of survival in many cancers including GBM.

C2. MTT assay is quick but the interpretation of the MTT data is limited. From an oncological point of view, clonogenic survival is a much better endpoint to evaluate treatment efficacy since it determines whether or not a tumor may relapse after end of therapy. Therefore, I strongly recommend that the authors confirm their key findings by determining clonogenic survival.

R2: It is a very valid suggestion. Actually, we have used 3D culture model in our previous investigation and showed that combination of Simva-TMZ disrupts tumor colony formation in 3D culture and significantly changed the structures (Shojaei et al 2020, FEBS J, Figure 16E). Therefore, we have not repeated this experiment in our current study.

C3. Provided that the tumor-eradicating efficacy of the combination therapy can somehow be deduced from the MTT data, the additional (additive) effect of statins (about 15% less viable cells as compared to temozolomide treatment alone) is not huge questioning a strategy to combine statins with temozolomide.

R3: We do agree that the combination effect won’t add more than 15 to 20% to the death of tumor cells but it is inhibiting autophagy flux which could be an important strategy against TMZ-chemoresistance in GBM. We have recently established TMZ resistant U251 cells and showed Simva-TMZ is able to initiate apoptosis in these cells and breaks chemo resistance (Shojaei et al, unpublished data).

Specific comments

Introduction

C4. Page 1, lines 60-61: “Epidemiological studies have shown statins reduce mortality rates of different types of cancer, including GBM [15,16].” Epidemological evidence for a statin use-mediated survival benefit is debated (J Clin Neurosci. 2017, 42:196-197). The reference #16 (in contrast to the statement of the authors, see Suppl. Figure S14 of this reference) and a more recent study (J Clin Neurosci, 31 (2016), pp. 176-180) do not show a survival benefit for brain tumor and glioblastoma patients, respectively, with statin prescription. Please, provide more detailed information.

R4. We appreciate these comments. We added a sentence and explained that although statin use decreases the mortality but it was not significant. Re: J Clin Neurosci, 2017, 42:196-197, the authors have indicated that “Due to the small number of patients and retrospective nature of the study, further work is needed to understand the role of perioperative statins in GBM patients”. Therefore it is still debatable and the results of ongoing clinical trial will provide more answers to this important question.

C5. Page 2, line 52: “with a median survival rate of less than 3% for 12−15 months [7-9].” This statement is not correct.

R5: We have corrected the sentence (red font, line 51).

C6. Page 2, lines 55, 56: “However, the survival chances of GBM patients remained low (~21 months) because of development of resistance to TMZ treatment 56 [9,13]” 21 months??? See sentence above. Please, introduce the temozolomide resistance gene MGMT and the MGMT promoter methylation in a subgroup of glioblastoma patients. By the way, standard therapy according to Stupp et al. comprises surgery, temozolomide radiochemotherapy and temozolomide maintenance chemotherapy.

R6: We have removed the sentence. As the focus of our manuscript is not on MGMT we have not introduced it to keep the introduction short and concise. We have added Stupp reference and modified “Since 2009, the gold standard therapeutic approach for GBM has been radiotherapy concomitant with temozolomide (TMZ) chemotherapy” to “GBM therapy comprises surgery, temozolomide radiochemotherapy and temozolomide maintenance chemotherapy” (line 52,53).

Results

C7.Page 2, line 93: Peak simvastatin plasma concentration (after oral dose of 40 mg Br J Clin Pharmacol. 2009,67:520–526) is in the range of 30 nM. Why do the authors apply micromolar concentrations?

R7: It is an important question. We agree that the plasma dose range of simvastatin is around 30 nM but we have to consider that the half-life of simvastatin in the media is very short and to keep an effective concentration of simvastatin for 72 h, we used micro molar concertation in our study (1, 2.5 uM) as per our previous studies (PMIDs: 31545550, 28344327)

Figure 1:

C8. The representative plots given are not very convincing. I cannot see the effects evident from the semiq-uantification of the bands (Figs. 5s and 6s). The authors do not describe and discuss the differences between the 0 h and 72 h control. Moreover, some plots are too dark. Some plots give a single band for U87 cells and a double band for U251. Please exchange the plots by more representative ones (Figs 5s,C and 6s,C: XBP-1s instead of XBP-1)

R8: We have provided the best representative blots and have corrected these wherever necessary. We used T:0 control to show the status of the event in the beginning of the treatment. Regarding the double band there might be different isoforms between two cell lines for each protein. We have corrected Figs 5s C and Fig 6s C. We have added a statement regarding control zero (line 129-130, page 3) and difference in some bands between U87 and U251 cells (line 136-136, page 3) in the revised manuscript.

Figure 2:

C9. This applies to all figures. Please, increase the fond size of the axis labeling.

R9: We apologize for this. We have corrected all figures for the font size in the revised manuscript.

C10. As discussed also by the authors in the discussion section, MTT assay measures NAD(P)H-dependent formation of formazan and is not the best surrogate marker for viability especially when interfering with autophagy. Why didn’t the author determine clonogenic survival by colony formation assay which is the gold standard in oncology?

R10: Like any assay MTT assay has its limitations which we have addressed in the MS. We have used this assay in past as a valid assay in studying cell death mechanisms as we have used in many of our previous investigations (Ghavami et al, 2004, J Leuk. Biol, Ghavami et al, 2008, J. Cell Mol Med, Ghavami et al, 2009, 2010, 2014, BBA-Mol Cell Res, Ghavami et al, 2010, Cell Res, Alizadeh et al, 2017, Sci Rep, Shojaei et al, 2020, FEBS J). We also used 3D culture model to evaluate the effect of Simva-TMZ treatment in 3D culture model, which showed the same results (Shojaei et al, 2020) using live/dead assay.

C11. the applied temozolomide concentration above the clinically achievable concentration in the tumor. Nevertheless, the effect of temozolomide on formazan formation is only some 20% (also shown in Fig. 1s and 2s) although U251 and U87MG cells reportedly are MGMT negative (Chin J Cancer. 2014,33:115–122) Please, discuss.

R11. We thanks reviewer for asking this clarification. We used 100 uM concertation of TMZ as TMZ is unstable in the media and we want to keep the TMZ in a constant level during the treatment (we added a sentence in the manuscript to describe it, line 102-104, page 3). The cytotoxic effect of TMZ is around 15-20% with this concertation in 72 hrs. The higher concertation of TMZ (>200 uM) has far more cytotoxic effect on both cell line. It shows the cytotoxic effect of TMZ is concertation depended and we have not discussed it in the current manuscript, as it is well described in several previous investigations.

C12. line 154 (and all further Legends). µ is mising

R12. We have added µ to all missing places and showed it with red font in the revised manuscript.

Figure 3s:

C13. subG1 population in Nicoletti staining is not specific for apoptotic cell death. This is also evident from the comparison of the subG1 population (Fig. 3s) and cleaved caspase-3 (Figs. 5s, F and 6s, F) data. Please reword (cell death instead of apoptosis), also throughout the main manuscript.

R13: We have changed “apoptosis“ to “cell death” in the revised manuscript (red font).

C14. Results, page 3, line 105: It is worth nothing

R14: We have changed “nothing” to mentioning”. It was a typo mistake (red font).

C15. Results page 3, line 111: lilipidated? lipid-conjugated

R15: “Lipidated” is very common in autophagy field (Autophagy guideline 2008, 2012, and 2016) and we have also used in our previous publications. Therefore, we keep it as our previous publications and autophagy guidelines.

C16. Results, page 6, lines 179ff and throughout the further text: “…we concluded that XBP-1 splicing is involved in the Simva-TMZ-mediated inhibition of autophagy flux, but that it does not affect the cytotoxic effects of Simva-TMZ”. , The “cytotoxic” effects are merely based on formazan formation which is a very week read-out (authors showed in Figs 2s and 3s discrepancies between formazan formation and DNA degradation). Please, confirm data with more conclusive endpoints such as colony formation assay.

R16: We agreed with the reviewer. To further expand these studies, we are currently doing investigation if inhibition of XBP splicing would affect tumor proliferation and migration via changing the secretome of GBM cells. We will address the current question in our next research publication in this field. In addition, the deadline for submission of abstract was only 10 days and it is not possible to do this experiment in short time of revision time.

C17. Figure 6: The pattern of the bands in figure part A is not consistent with the bar diagrams shown in B and C. E.g., the high abundances of p-eIF2alpha in U87 cells upon PERK inhibition + temozolomide is not represented by the p-eIF2alpha/eIF2alpha ratio. The latter is similar to PERK inhibition + simvastatin which shows only a weak band of p-eIF2alpha in A. Please, give more representative blots.

R17: We have included the best representative blots of our results. As we have normalized all ratios to GAPDH in each band, the densitometry results could be seen different.

Discussion

C18. page 14, line 334 (and throughout the text): “… IRE-1 expression” better protein amount or abundance

R19: We have changed “protein expression” to “protein amount” in all related place in the revised manuscript (red font).

C19. page 14, line 336ff “…Simva-TMZ elevated cleaved caspase-3 and p-H2A.X (an indication of double-strand 336 breaks) more than TMZ alone, especially in U251 cells. This implies that the effect of Simva-TMZ on.” data on gammaH2AX are not described in the result section and don not seem very robust. In U87, simvastatin/temozolomide co-treatment reduces gammaH2AX abundance (Fig. 5s) and how do the authors explain the increase in gammaH2AX abundance just by 72 h cell culture?

R19: Actually, we have included gamma H2X data to confirm the effect of TMZ in the model. As the focus of our paper was not on DNA repair and its connection to UPR and autophagy we have not focused on gamma H2X abundance and its difference between single TMZ, and Simva or Simva-TMZ combination therapy.

Material and Methods

C20. Page 17, line 448: 5% CO2 - DMEM medium usually contains 3.4 g/l bicarbonate and requires higher CO2 concentrations to equilibrate pH 7.4.

R20: We usually checked the pH of media in 5% CO2 and the pH was always 7.4. Based on the manufacture and ATCC recommendation, the CO2 pressure could be between 5 to 10% to maintain the pH and we used 5% and it was working for our experiments.

Round 2

Reviewer 1 Report

The authors have responded to the comments. 

Reviewer 2 Report

The authors addressed the most of my concerns and so, about my opinion, the work is now suitable for publication in Cancers journal.

Reviewer 3 Report

Dear authors,

this paper has been improved only marginally during the revision process. Beyond commenting in the rebuttal letter, most of my suggestions were not implemented in the manuscript. The absurd deadline for re-submission given by the editorial team is - in my opinion - not an excuse to not consider additional experiments since everybody will understand that you cannot do major revisions within 10 days and will grant extension upon request.

Still, wrong statements (e.g., lines 353ff "In addition, Simva-TMZ elevated cleaved caspase-3 and p-H2A.X  more than TMZ alone, especially in U251 cells.") are given in the revised manuscript  (in fact, only in U251, Simva-TMZ co-incubation increases  p-H2A.X protein abundance, in U87, the opposite occurs. In the blots (Fig .1, Fig. 4s, Fig. 5s, Fig. 6s), there is a strong effect of  incubation time in the control condition on the abundance of several proteins which in some experiments is extremly significant. This effects were not commented suggesting that the authors do not have any hypothesis that might explain these effects which casts severe doubts on robustness of the observed inibitors-mediated effects.

Another point: you cannot study Temozolomide effects in glioma without dealing with MGMT